# Impact of Activity-Oriented Propioceptive Antiedema Therapy on the Health-Related Quality of Life of Women with Upper-Limb Lymphedema Secondary to Breast Cancer—A Randomized Clinical Trial

**DOI:** 10.3390/jcm11071884

**Published:** 2022-03-28

**Authors:** María Nieves Muñoz-Alcaraz, Luis A. Pérula-de Torres, Antonio José Jiménez-Vílchez, Paula Rodríguez-Fernández, María Victoria Olmo-Carmona, María Teresa Muñoz-García, Presentación Jorge-Gutiérrez, Jesús Serrano-Merino, Esperanza Romero-Rodríguez, Lorena Rodríguez-Elena, Raquel Refusta-Ainaga, María Pilar Lahoz-Sánchez, Belén Miró-Palacios, Mayra Medrano-Cid, Rosa Magallón-Botaya, Mirian Santamaría-Peláez, Luis A. Mínguez-Mínguez, Jerónimo J. González-Bernal

**Affiliations:** 1Inter-Level Clinical Management Unit of Physical Medicine and Rehabilitation, Reina Sofía University Hospital—Córdoba and Guadalquivir Health District, Andalusia Health Service, 14004 Córdoba, Spain; marian.munoz.sspa@juntadeandalucia.es (M.N.M.-A.); maviol@hotmail.es (M.V.O.-C.); mariat.munoz.garcia.sspa@juntadeandalucia.es (M.T.M.-G.); chonjorge@hotmail.com (P.J.-G.); 2Maimonides Institute for Biomedical Research of Córdoba, Reina Sofía University Hospital, University of Córdoba, 14011 Córdoba, Spain; langel.perula.sspa@juntadeandalucia.es (L.A.P.-d.T.); jesussleep@hotmail.com (J.S.-M.); espe_mrr@hotmail.com (E.R.-R.); 3Multiprofessional Teaching Unit for Family and Community Care of the Córdoba and Guadalquivir Health District, 14011 Córdoba, Spain; 4Valle de los Pedroches Hospital, Andalusia Health Service, 14400 Pozoblanco, Spain; jimenezvilchez14@gmail.com; 5Department of Health Sciences, University of Burgos, 09001 Burgos, Spain; mspealez@ubu.es (M.S.-P.); jejavier@ubu.es (J.J.G.-B.); 6San Jose Health Center, Aragonese Health Service, 50013 Zaragoza, Spain; anerol.er@gmail.com (L.R.-E.); minirefus@hotmail.com (R.R.-A.); pilarlahoz95@gmail.com (M.P.L.-S.); 7Association of People with Lymphedema in Aragon (ADPLA), 50007 Zaragoza, Spain; adplaragon@hotmail.com; 8Lozano Blesa University Clinical Hospital, Aragonese Health Service, 50009 Zaragoza, Spain; medranocid@hotmail.com; 9Institute for Health Research Aragon (IIS Aragon), University of Zaragoza, 50009 Zaragoza, Spain; med000764@gmail.com; 10Department of Educational Sciences, University of Burgos, 09001 Burgos, Spain; laminguez@ubu.es

**Keywords:** occupational therapy, lymphedema, breast cancer, quality of life related to health, complex decongestant therapy, activity-oriented proprioceptive antiedema therapy

## Abstract

Background: Alterations derived from lymphedema in the upper-limb secondary to breast cancer-related lymphedema (BCRL) decrease the health-related quality of life (HRQoL), but there is limited evidence of the impact of the different interventions on it. The aim of this research was to compare the effect of conventional treatment with another treatment based on Activity-Oriented Antiedema Proprioceptive Therapy (TAPA) on HRQoL in women diagnosed with BCRL. Methods: A prospective clinical study was designed with two parallel arms. The study population consisted of women diagnosed with BCRL in stage I and II, belonging to different institutions in Córdoba and Aragon, Spain. Sociodemographic and HRQoL-related variables, pain, tightness, heaviness and functionality were obtained before and after treatments. Results: 51 women participated in the study, 25 received the conventional treatment and 26 the TAPA, with a mean age of 59.24 ± 9.55 years. HRQoL was significantly related to upper-limb function and pain on the participants’ affected side. In addition, covariance analysis (ANCOVA) showed that the TAPA treatment interfered less in the performance of activities of daily life and produced significant improvements in the social dimension of HRQoL. Conclusions: the non-use of compressive elements in the rehabilitative treatment of the BCRL that is proposed with TAPA improves aspects such as self-image and participation in social and recreational activities.

## 1. Introduction

Upper-limb lymphedema secondary to breast cancer (BCRL) is the accumulation of fluid in the interstitial space, resulting from damage to lymph nodes during axillary dissection and radiation. It is a chronic and progressive condition, in which insufficient drainage of lymphatic fluid causes swelling and alterations in the upper extremity on the side of the breast in treatment and in functions related to skin, energy and impulses, sexuality, sleep, temperament and personality, as well as in additional sensory functions such as proprioceptive and tactile functions. All these alterations limit the performance of activities and restrict participation in daily situations, decreasing the health-related quality of life (HRQoL) of people suffering from this disorder [1,2,3,4].

The main goals in lymphedema treatment are to limit morbidity and to improve patient functionality and HRQoL. BCRL is still a challenging condition either for breast cancer survivors and clinicians, deeply impacting patient functioning and quality of life. Due to the lack of globally accepted criteria in evaluating, to date, a gold-standard treatment for this widespread issue is still needed, and some authors present the recommendations on interventions developed from the evidence review, according to stage, to allow for clinical implementation based on patient presentation [5,6]. A task force of the Academy of Physical Therapy Oncology of the American Physical Therapy Association (APTA) developed a clinical practice guideline [6], recommending with a level A, for stage I education, compression garments and an exercise program, and for stage II, compression bandaging and exercise. Symptom management can be conservative or surgical, HRQoL is often considered in the choice of the therapeutic modality [7,8]. The standard conservative treatment is Complete Decongestive Therapy (CDT) or Combined Physical Therapy (CPT), which consists of a combination of techniques that includes skin care, manual lymphatic drainage, joint mobility exercises, compression garments and multilayer bandages; however, there are no solid data to recommend its universal use due to the characteristics of the treatment itself [9,10,11]. The prescription of compression garments for lymphedema is very varied due to the lack of evidence to support the treatment [12], whose use neither provides benefit nor is contraindicated during the practice of physical activity [13], and it can cause irritation, skin discomfort and pain, and even soft tissue and nerve injuries [14,15]. In addition, there is no optimal level of adherence to its use as it causes patients discomfort and negative emotions related to their visibility and interference with function and social situations [16]. Although the prescription of compression garments for lymphedema is highly varied and may be due to a lack of supporting evidence to inform treatment [16], the use of light compression sleeves for 2 years may reduce the incidence of lymphedema, and lead to significant improvement in important quality-of-life parameters, such as physical functioning, fatigue, arm pain and breast symptoms [17].

In this sense, patient-reported outcome measures provide valuable information with greater prediction about the effect of BCRL on their HRQoL, rather than objective measurements, and may guide therapeutic decisions, but their heterogeneity hinders international research efforts to improve treatment methods and HRQoL for patients with lymphedema [17]. Current evidence makes visible the need for further research in cancer rehabilitation, and recommends alternative techniques design and implementation more focused on psychosocial and quality-of-life factors regarding the management of symptoms in patients with lymphedema [4].

Based on the previous statements, this research hypothesises that better results can be obtained in terms of satisfaction, comfort and empowerment perceived by patients by replacing compression by occupation in the treatment of BCRL. The objective of this research was to compare the effect of CDT with another experimental conservative treatment based on Activity-Oriented Antiedema Proprioceptive Therapy (TAPA) on generic and specific HRQoL of women diagnosed with BCRL in stage I and II and analyze possible associations between the symptoms perceived by patients in the affected arm and their health-related quality of life.

## 2. Materials and Methods

### 2.1. Study Design

A controlled, multicentric, prospective, stratification-randomised clinical trial was designed in two gradients, single-blind and with two parallel arms: a control group (CG) in which the rehabilitation guidelines recommended in the Integrated Breast Cancer Care Process of the Ministry of Health and Families of the Junta de Andalucía [18] were followed (preventive measures, skin care and exercise, prescription of compression garments and multilayer bandages and manual lymphatic drainage, using compression garments, in both stages, in the maintenance phase, after the intensive treatment) and an experimental group (EG) that received the TAPA treatment, which does not exert compression on the affected upper-limb and uses activity as a treatment method (Figure 1).

The experimental group underwent TAPA treatment in stages I and II. The participants received 10 sessions (2 weekly), of 30 min each, directed by two occupational therapists.

The intervention was based on therapeutic activity for the reduction of the volume of lymphedema, which is significant for each person and whose graduation includes:-Activities with neurodynamic slip patterns;-Proprioceptive neuromuscular facilitation activities;-Proprioceptive cohesive anti-edema bandage, with a technique similar to the Coban-type bandage, but without compression and with a high cotton content. The patient and/or caregiver is instructed on its use and placement and modifications or adaptations are recommended for optimal performance in their ADLs.

At the end of the 10 sessions, each patient performed 5 individually prescribed activities daily and did not use any compression garments in its maintenance phase.

The study protocol was previously registered on the ClinicalTrials.gov website of the U.S. National Library of Medicine, with reference number NCT03762044, and published in the peer-reviewed open access journal BMC Cancer [19]. 

The results reported in this manuscript refer to the quality-of-life levels of women with BCRL who receive conventional treatment or TAPA, as the main variables analysed in this study.

### 2.2. Study Participants and Recruitment

The study population consisted of women surgically treated for breast cancer (BC) and diagnosed with BCRL in stage I and II according to the ISL 2020 lymphedema scale [10], belonging to the Inter-level Rehabilitation Clinical Management Unit of the Reina Sofía University Hospital in Córdoba, the Association of Aragonese Women with Genital and Breast Cancer (AMAC-GEMA), the Association of People with Lymphedema of Aragon (ADPLA) and/or the San José Health Centre and the University Clinical Hospital of Aragon.

Women with diseases or dysfunctions incompatible with treatment, with comorbid pathologies that could skew the research’ results or with bilateral lymphedema were excluded.

Providing signed informed consent was an essential requirement to participate in the study.

### 2.3. Sample Size

The literature estimates a minimum detectable volume value of lymphedema at a difference of 2.39% (42.9 mL) from baseline [18]. Considering means and standard deviations from other studies [20] for an alpha error of 0.05 and a statistical power of 80%, a sample size of 29 subjects per group is calculated with EPIDAT 4.2. The effect of standard treatment in the control group is estimated at a reduction in mean arm volume of 5% and 20% for experimental and similar standard deviation in both groups, close to 20% [21]. Taking into account a dropout rate of 10% in each group, the estimated corrected sample size is 64 patients, randomly assigned to two groups of 32 patients each, 16 per stage and intervention group. The final sample consisted of 51 patients (Figure 1).

### 2.4. Procedure and Randomisation

The participants were selected through a consecutive sampling, with the women from the participating institutions who met the inclusion criteria being invited to participate in the study as they were identified and recruited by the field researcher in charge of this task. The sequence of random assignment of the participants to be included in the GC or EG was carried out with the statistical software EPIDAT, 3.1 (https://www.sergas.es/Saude-publica/Epidat-3-1-descargar-Epidat-3-1, accessed on 23 March 2022), stratifying patients by stage of lymphedema I or II, according to the classification of the International Society of Lymphology, with a ratio of 1:1, using blocks of random size of 4. It was concealed from the investigator who evaluated the participants using sequentially listed envelopes that were opaque, sealed and stapled. The assignment was made by the people in charge of contact and recruitment.

Prior to data collection and treatment development, all study participants attended a group lymphedema health education workshop. The workshop lasted 3 h and was conducted by the people in charge of recruitment, who taught basic knowledge about the pathophysiology of lymphedema, the early identification of symptoms, preventive skin care measures, guidance for ADL performance, and various exercise guidelines and anti-edema postures.

A procedure manual was designed for data collection, which facilitated the project’s correct development, as well as a notebook with all the sociodemographic and clinical variables of interest obtained through structured interviews and data from the medical records. Data were collected by researchers previously trained for this task, who performed a first evaluation during the 24–72 h before starting assigned treatment (pre-test) and a second evaluation during the 24–72 h after the end of treatment (post-test).

To minimise cross-contamination between groups, both the evaluators and the researchers in charge of monitoring and analysing the data remained blind to the group to which the participants belonged. At the same time, all participants were clearly instructed not to disclose to blinded investigators during assessment visits the group to which they had been randomly assigned.

The project was approved by the Research Ethics Committee of Córdoba, in a meeting held on 27 November 2018 (Act nº 282, ref. 4084) and with the authorisation of the Management/Direction of the Health District of Córdoba and Guadalquivir, the Reina Sofía University Hospital and the AECC Headquarters Córdoba.

### 2.5. Main Outcomes

The HRQoL variable was analysed using the Upper Limb Lymphedema 27 Value (ULL-27). This is a specific instrument to measure the quality of life in patients with BCRL, adapted cross-culturally and validated in its Spanish version in 2016 (the result is calculated as a percentage of the social, physical and psychological dimensions) [22]. Its results indicate that the higher the score obtained, the greater the disability or the worse the symptomatology of HRQoL. Each item is scored according to a 5-point Likert scale and the patient self-assesses the frequency with which they felt difficulty performing the activities and/or had the feelings defined in the scale, in which 1 means “never”, 2—“very few times”, 3—“sometimes”, 4—“many times” and 5—“always”. The resulting quotient is multiplied by 100 in order to obtain a result from 0 to 100 for each dimension (0 corresponds to the best possible HRQoL and 100 to the worst) [22]. In addition, the value of the EUROQOL thermometer of health status’ Self-Assessment of the Health Questionnaire of EuroQol Group (EQol-5D) was used, a generic, standardised and validated instrument in Spain, which is used to describe and assess the HRQoL of a group or population. EQol-5D is a vertical visual analogue scale (VAS) of 20 cm, millimeter, ranging from 0 (worst imaginable state of health) to 100 (best imaginable state of health). In this case, the person must mark the point on the vertical line that best reflects the assessment of their state of global health on the given day. A higher score on this questionnaire indicates a worse HRQoL [23].

The VAS was used to assess the sensations of pain, heaviness and tightness perceived by the patients in the upper limb. This scale is represented by a continuous horizontal line of 100 mm, whose value 0 (on the left) indicates their absence and 10 (on the far right) indicates their extreme presence—the highest level of gradation [24].

For the evaluation of performance, the Brief Disability of the Arm, Shoulder and Hand Questionnaire (Quick DASH) was applied, which evaluates the ability to perform 11 activities during the past week, with answer options from 0 to 5, where 0 indicates “nothing” and 5—“extremely”, being able to reach values from 0 (absence of difficulty) to 100 (severe difficulty) [25].

In all outcomes, higher differential scores show better results.

### 2.6. Statistical Analysis

To minimise and control the effects of non-random losses and dropouts, an intention-to-treat analysis was performed. In the descriptive analysis of the sample, mean and standard deviation (SD) were used in the case of quantitative variables, or absolute frequencies and percentages for categorical variables. Compliance with the normality criteria in the quantitative variables was verified using the Kolmogorov–Smirnov test. Before proceeding to the statistical analysis, the differential score (DS) of all the continuous variables was calculated, subtracting the pre-test or pre-intervention score from the post-test or post-intervention score (post-test − pre-test), in order to analyse the variation in the score once the intervention was finished and not only the post-treatment score. To evaluate the relationship between the different dimensions of quality of life and the rest of the continuous variables, the Pearson correlation test was used.

Covariance analyses (ANCOVA) were performed to compare the differences in the different dimensions of quality of life between the GC and EG, using the pre-tests of the dependent variables as covariates and the intervention groups as a fixed factor. The effect size of the interventions was estimated using the eta squared coefficient (η^2^), interpreted according to the following criteria: if 0 ≤ η^2^ < 0.05, no effect; if 0.05 ≤ η^2^ < 0.26, the effect was minimal; if 0.26 ≤ η^2^ < 0.64, the effect was moderate; and if η^2^ ≥ 0.64, the effect was strong [26].

Statistical analyses were performed with SPSS software version 25.0 (IBM SPSS Inc., Chicago, IL, USA). Statistical significance was considered if *p* < 0.05.

## 3. Results

### 3.1. Main Characteristics of the Participants

The study sample was composed of 63 women, of whom 32 were assigned to the CG and 31 to the EG. There were 12 losses in the study, either due to an inadequate level of adherence to treatment or due to problems in data collection, so finally, 51 participants completed the study and were included in the analysis, 25 in the CG (15 patients in stage I and 10 in stage II) and 26 in the EG (15 in stage I and 11 in stage II) (Figure 1).

Table 1 summarises some of the main characteristics of the participants according to the study group. The mean age was 59.24 years (SD ± 9.55), and most women were active (*n* = 26; 51%) or retired (*n* = 19; 37%). Only two participants (4%) were able to undergo breast-conserving surgery and in the remaining 49 (96%), surgical treatment was by mastectomy. Most participants had no complications after surgery (*n* = 43; 84%).

### 3.2. Quality of Life: Pain, Heaviness, Tightness and Performance

Table 2 summarises the relationships between the different dimensions and measures of quality of life and the variation in the levels of pain, heaviness, tightness and upper-limb functionality on the affected side. After calculating the differential scores of the study variables (post-test − pre-test) to obtain the variation in the results at performance of the upper limb on the affected side, after the end of the intervention in both groups, the Pearson correlation test revealed a positive relationship between the social dimension and performance (*p* = 0.037), which suggests that the more the functional capacity of the upper limb of the affected side improves during the intervention, the more it increases its HRQoL in the social dimension and vice versa.

HRQoL was also significantly correlated with pain (*p* = 0.048) and performance (*p* = 0.004), so the greater the improvement in the pain variable and in the functional capacity of the upper limb of the affected side during the intervention, the greater the increase in HRQoL, and vice versa (Table 2).

### 3.3. HRQoL: Differences between Groups

ANCOVA showed statistically significant differences between GC and GE in the social dimension of the ULL-27 scale controlling the scores obtained in the pre-test, so those differences could be attributed to the intervention performed. This means that HRQoL, in its social dimension, improved more in the group of participants who received the TAPA treatment, based on activity as a treatment method and without compression in the upper limb of the affected side, compared to the group that received conventional treatment (Table 3). Despite being significant, the effect size was (η^2^ ≤ 0.076), and no statistically significant differences were obtained in the other HRQoL dimensions studied, and neither in the measurement carried out at three months. The results of this study do not show significant differences in the volume reduction of BCRL between intervention groups (control and experimental). There are no significant differences in volume in the pretest between the groups (U = 369.00; *p* = 0.407), although there are significant differences between the pre-test and post-test of both groups, that is, all patients significantly reduced volume (Z = 463.00, *p* = 0.61).

The following graphs represent the means of the pre-test and post-test of the variable with statistically significant differences between groups at post-test (Figure 2 and Figure 3).

## 4. Discussion

This study measures the effect of the application of TAPA on HRQoL, allowing BCRL patients to self-assess their own state of health, in its physical, social and psychological dimensions, as well as upper-limb functions and perceived sensations of pain, heaviness and tightness, using instruments with strong psychometric properties that allow to provide information scarcely documented in the evidence, but very necessary both for therapeutic decision making by health professionals and lymphedema patients and their caregivers.

Related to this, previous systematic reviews [17,27] support the need to adopt a holistic approach and consider psychosocial, clinical and sociodemographic variables, similar to the approach in this clinical trial, in order to better understand the HRQoL of these patients and implement future appropriate preventive measures.

In accordance with the results obtained by Togawa et al. [28], this study shows how perceived symptoms of pain, heaviness and tightness negatively impact HRQoL and occupational performance. At the same time, as Bojinović-Rodić et al. maintain [29], our results also show that performance difficulties significantly influence a worse perception of HRQoL.

In the analysis of results on the effect of both treatments on HRQoL, the social dimension of HRQoL on the ULL-27 scale was further improved in the group of participants who received the experimental treatment, TAPA. This social dimension in which there was more improvement includes positive effects on self-image, reduction of difficulties to enjoy outdoor life, participate in personal projects, maintain an effective life and visit leisure establishments. The impact of lymphedema on the performance and participation in leisure and recreation activities, as well as the need for support and social relations, is not documented in the literature prior to 2014 [2], despite the fact that patients report difficulty in all categories of activity included in the International Classification of Functioning, Disability and Health (ICF) of the World Health Organisation [30], mentioning first elements of mobility, followed by personal care, recreation and leisure and domestic life. This negative association between restrictions on participation and difficulties in carrying out activities and HRQoL was also studied by Nascimento et al. [31], who determined the need to focus therapeutic strategies on restoring occupational performance, which was the approach of this study. It was shown that patients consider a burden the use of compression garments and bandages, perceiving that the use of this environmental factor negatively affects their HRQoL [2]. This discomfort of compression garments stated by patients is also addressed by Al Onazil et al. [32], who point to the interference with the function and visibility of the garment among the reasons that lead to poor adhesion to its use. Not using compression garments with the experimental treatment proposed in this study could be one of the justifications that support the results obtained, with a better participation of patients in the social dimension. A lack of consistent benefit or occurrence of any adverse effect of wearing a compression garment in performing physical activity is also the conclusion of the systematic review conducted by Hayes et al. [14].

Physical exercise is a planned, structured and repetitive activity and TAPA proposes the prescription of significant activity as a means to improve HRQoL. This therapeutic potential of physical activity found in our results coincides with the findings of Baumann et al. in their systematic review [33], where they report additional improvements in the results reported by patients with BCRL in HRQoL and mood with physical exercise. Additionally, many authors support the benefits of physical activity to obtain substantial improvements in HRQoL, but also of all aspects of the evolution of patients diagnosed with BC, not only those with BCRL [34,35,36]. Parisa Mokhtari-Hessari and Ali Montazer found in their review [37] that physical activity and psychosocial interventions have been shown to be effective in improving the quality of life of this population. Other interventions that improve HRQoL in patients with lymphedema in the upper limb are also found in the literature, such as the study by Marzia Salgarello et al. [38], who used lymphaticovenular anastomosis, also observing with this procedure, a reduction in episodes of lymphangitis, and a decreased need for conservative treatment. However, as described by Morgan L. Fish et al. in their systematic review [7], there are few studies that evaluate the results for health-related quality of life, for the different therapeutic modalities available. Four CDT studies in this review reported improvement in all HRQoL subscales, but three studies reported no significant improvement in HRQoL after CDT, of which two had long-term follow-up.

This study proposes the viability of the intervention presented, TAPA, in different health contexts and at different levels of relevant care (primary and specialised health care) with the multi-centre nature of the clinical trial, promoting the multidisciplinary intervention of a disorder for which the need for a biopsychosocial approach has been documented and thus also providing external validity to the results obtained. The study offers a therapeutic alternative for a complex disorder, still without definitive treatment, and for whose prevention there is controversy regarding the choice of the most appropriate interventions [39], and which improves HRQoL, affected in all its dimensions in people who suffer from it, eliminating possible adverse effects and/or rejection of compressive elements and providing another treatment option. Experimental treatment with AAPT brings better benefits in aspects such as self-image and participation in social and recreational activities than conventional treatment. Presumably, the less physically demanding TAPA may not be perceived as ‘better’ if the patient is influenced by the traditional use of CDT.

Regarding its limitations, the scarce evidence available on the effectiveness of eliminating compression garments in the conservative treatment of BCRL, especially in its maintenance phase, makes it difficult to discuss and compare the results, so further research will be needed to address this aspect with a larger sample of the population. Since the study participants were women diagnosed with stage I and II of BCRL and the context of the intervention was individual, its effect on stages 0 and III, in men and in group settings is unknown.

## 5. Conclusions

The experimental treatment with TAPA may provide additional benefits in the social dimension of the HRQoL of patients with BCRL, compared to the conservative approach. Likewise, statistically significant relationships were found between HRQoL and pain, heaviness, tightness and performance of the upper limb on the affected side of the participants. This study suggests that the non-use of compressive elements in the rehabilitative treatment of BCRL proposed with AAPT can improve aspects such as self-image and participation in social and recreational activities. More research is needed to observe the long-term maintenance of the effect in the reduction of lymphedema volume, and its impact on the health-related quality of life of people with BCRL.

## Figures and Tables

**Figure 1 jcm-11-01884-f001:**
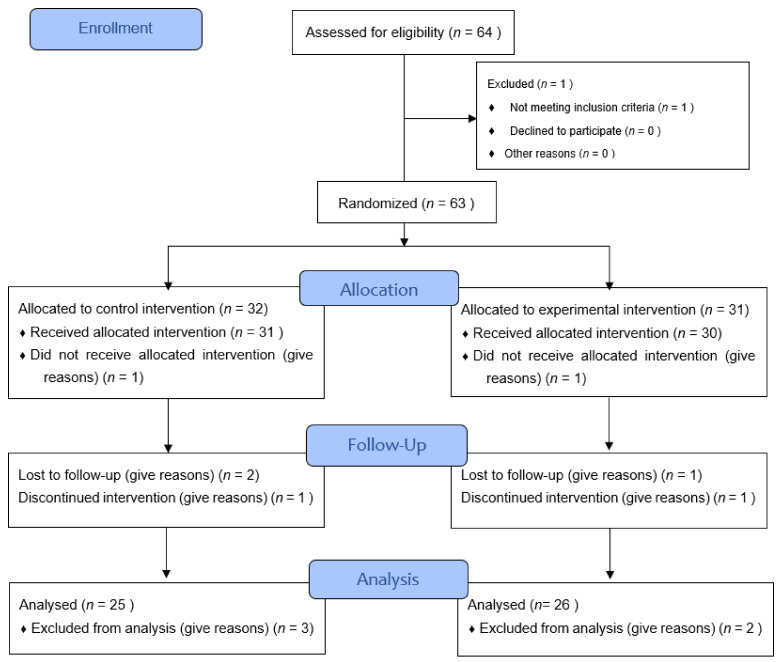
CONSORT flow diagram.

**Figure 2 jcm-11-01884-f002:**
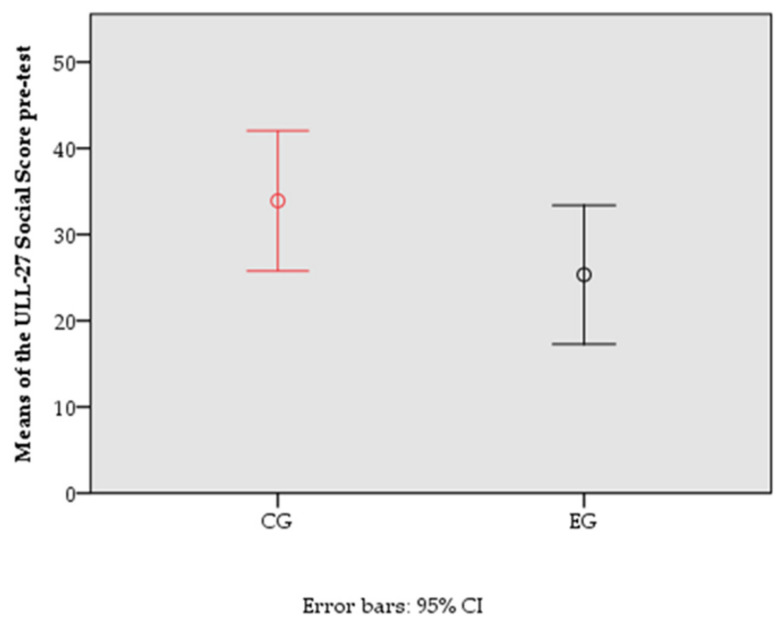
Means of the ULL-27 Social Score of the different groups in the pre-test.

**Figure 3 jcm-11-01884-f003:**
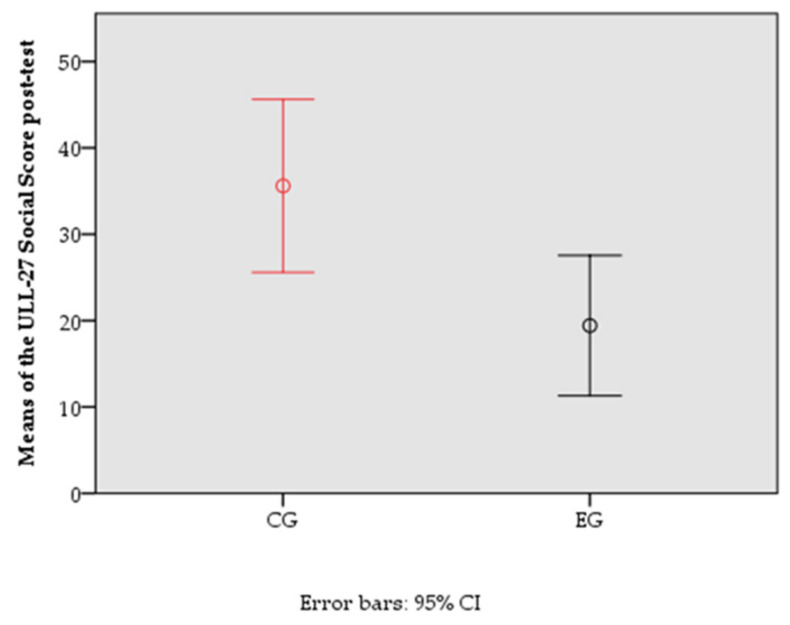
Means of the ULL-27 Social Score of the different groups in the post-test.

**Table 1 jcm-11-01884-t001:** Main characteristics of participants.

Variables	Total (*n* = 51)	CG (*n* = 25)	EG (*n* = 26)	*p*
**Age** (mean ± SD)	59.24 (±9.55)	61.80 (±9.83)	56.77 (±8.76)	
**Occupation**				
Active	26 (51%)	8 (32%)	18 (69%)	
Unemployed	4 (8%)	3 (12%)	1 (4%)	
Retired	19 (37%)	12 (48%)	7 (27%)	
Other	2 (4%)	2 (8%)	0 (0%)	
**Type of treatment received**				
Conservative	2 (4%)	1 (4%)	1 (4%)	
Mastectomy	49 (96%)	24 (96%)	25 (96%)	
**Complications**				
No	43 (84%)	21 (84%)	22 (84%)	
Infection	3 (6%)	2 (8%)	1 (4%)	
Seroma	4 (8%)	1 (4%)	3 (12%)	
Keloid scar	1 (2%)	1 (4%)	0 (0%)	
HRQoL		65.40	66.54	0.822
QuickDASH		46.88	37.92	0.162
ULL-27 physical		48.54	41.75	0.250
ULL-27 social		23.09	24.04	0.137
ULL-27 psychological		48.76	45.89	0.504
VAS		4.04	2.65	0.063

Quality of life related to health (HRQoL); The shortened Disabilities of the Arm, Shoulder and Hand questionnaire (QuickDASH); Upper Limb Lymphedema (ULL-27); Visual Analytical Scale (VAS). Values are expressed as mean ± standard deviation (SD) or relative frequencies (percentages). CG: control group; EG: experimental group.

**Table 2 jcm-11-01884-t002:** Relationship between the different dimensions of HRQoL and pain, heaviness, tightness and performance of the upper limb on the affected side, using Pearson’s correlation.

	Differential ULL-27-Ph Score		Differential ULL-27-S Score		Differential ULL-27-Ps Score		Differential EQol-5D Score	
	r	*p*-Value	r	*p*-Value	r	*p*-Value	r	*p*-Value
**Differential VAS-P score**	0.135	0.346	0.252	0.074	0.173	0.225	0.279	*0.048*
**Differential VAS-H score**	0.049	0.733	0.247	0.052	0.037	0.794	0.247	0.080
**Differential VAS-T score**	0.318	*0.023*	0.256	0.070	0.185	0.193	0.154	0.282
**Differential Q-DASH score**	0.021	0.886	0.293	*0.037*	0.266	0.059	0.393	*0.004*

ULL-27-Ph: ULL-27 Physical; ULL-27-S; ULL-27 Social; ULL-27 Ps: ULL-27 Psychological; VAS-P: VAS-Pain; VAS-H: VAS—heaviness; VAS-T: VAS—tightness; Q-DASH: Quick-Dash; EuroQol-5D: EQol-5D.

**Table 3 jcm-11-01884-t003:** Comparison between groups in differential (post-test − pre-test) ULL-27 and EQol-5D scores, controlling pre-test scores, using ANCOVA.

Variable	Source	Type III Sum of Square	df	MS	F	*p*-Value	η^2^
Differential ULL-27-Ph score	ULL-27-Ph pre-test	1,802,313,482	1	1,802,313,482.445	3,406,908.005	<0.001	1
	CG/EG	1573.655	1	1573.655	2.974	0.091	0.058
	Error	25,392.833	48	529.017			
Differential ULL-27-S score	ULL-27-S pre-test	1119.004	1	1119.004	4.928	0.031	0.093
	CG/EG	1021.240	1	1021.240	4.497	*0.039*	0.085
	Error	10,898.700	48	227.056			
Differential ULL-27-Ps score	ULL-27-Ps pre-test	970.213	1	970.213	10.145	0.002	0.174
	CG/EG	64.764	1	64.764	0.677	0.414	0.013
	Error	4590.119	48	95.627			
Differential EQol-5D score	EQ-5D-5L pre-test	628.345	1	628.345	3.920	0.053	0.075
	CG/EG	122.982	1	122.982	0.767	0.385	0.015
	Error	7692.455	48	160.259			

ULL-27-Ph: ULL-27 Physical; ULL-27-S; ULL-27 Social; ULL-27 Ps: ULL-27 Psychological; MS: mean square; CG: control group; EG: experimental group.

## Data Availability

Not applicable.

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
