# Peer review of "Impact of Activity-Oriented Propioceptive Antiedema Therapy on the Health-Related Quality of Life of Women with Upper-Limb Lymphedema Secondary to Breast Cancer—A Randomized Clinical Trial"

_jcm, 2022, doi:10.3390/jcm11071884_

Round 1

Reviewer 1 Report

The authors have carried out a great job of coordination between different cities and have taken the trouble to make a good design. However, the article needs to be modified, since in its current form it is not aimed at responding to the objective that it poses. Please, review my comments to reword the objectives in a way that encompasses all the variables that have been studied, details the methodology more, and presents the results in a clearer way that responds to the objectives. I hope that my comments will help you to get the full potential of this article.

ABSTRACT:

Please, note that the abbreviation BCRL is for breast cancer-related lymphedema, so these words should precede that abbreviation

“of the person that suffers it” can be omitted

Author stay that evidence for the decrease the health-related quality of life in patients suffering BCRL is limited. It has been quite studied. I think they mean the evidence of the AAPT on HQRL is limited. Please rewrite accordingly. 

INTRODUCTION

“Lymphedema clinical practice guidelines  show a limited number of guidelines, few contemporary references, and low-quality studies”[5,6]- References 5 and 6 are not specific from BCRL, bur lymphedema in general. In fact, the American Physical Therapy Association classify interventions for BCRL by stage (including stages I and II) and evidence grades. Please see reference (doi: 10.1097/01.REO.0000000000000223). Compression garment, a tailored exercise program and education are recommended as first-line treatment with evidence Grade A (High-quality studies, level I) for BCRL at state I and II. 

Activity-Oriented Antiedema Proprioceptive Therapy, the main intervention of the study, is not explained in the introduction section, 

METHODS

“adapted cross-culturally and validated in its Spanish version in 2016”. Please add reference 

Is the VAS validated to measure heaviness and tightness perceived by the patients with BCRL?

If the objective of the research was to compare the effect of two therapies on HRQoL, why did the authors measure performance, sensations of pain, heaviness and tightness?

Regarding quick DASH, please add the reference of the spanish version and its validation in patients with BCRL instead of a website. 

How was each of the therapies carried out? how often and where?

 Please, add sample size. 

RESULTS

“Table 2 summarises the relationships between the different dimensions and measures of quality of life and the variation in the levels of pain, heaviness, tightness, and upper limb functionality on the affected side” QuickDASH measures functionality, and do not differentiate between affected side. 

Please, calculate the effect side of both interventions for EQol-5D and ULL-27 , so the objective of the study can be achieved (“to compare the effect of CDT with another experimental conservative treatment based on Activity-Oriented Antiedema Proprioceptive Therapy  on generic and specific HRQoL”

Please, add to table 1 baseline descriptive outcomes related to HRQoL (Qol-5D and ULL-27), function (quickDASH) and VAS and test if there is differences between groups at baseline. 

What is “differential ULL-27 scores” in table 3?

The names of figures 2 and 3 are not in English.  

If there was significant differences between groups in HRQoL at baseline (figure 2), results obtained from the present study can be explained by the intervention.

DISCUSION

“The study offers a therapeutic alternative to a complex disorder that, although it 320 is currently not possible to prevent or cure”… There are current guidelines which stay strength exercise to prevent BCRL 

“especially in its  maintenance phase” where women from this sample in maintenance phase? Did they received the same treatment in both groups previously? This is not reported in methods.

CONCLUSIONS

“AAPT can improve aspects such as self-image and participation in social and recreational activities”. Where is the data that supports this statement?

Reviewer 2 Report

Muñoz-Alcaraz and colleagues present data from a lymphedema clinical trial in which Activity-Oriented Propioceptive Antiedema Therapy is utilized in place of (control) compression strategies and assess quality of life surveys as a potential differential outcome. The overall effect, while significant, is minor. The introduction to the material, methodology of the study and statistical analyses, and discussion are all well-written and easy to follow.

As this outcome is potentially of great interest to therapists, researchers, and even patients, the results presentation needs to be improved. The heavy use of acronyms makes the tables and text unclear. Ideally, the authors could expand upon the terminology utilized to introduce how the study was performed (i.e., which surveys) and how then is it is compared, increasing the text in the Results reporting to put the Pearson values and the variable v. source in better context.

Would any lymphedema outcome factors be considered? Presumably the less physically demanding AAPT may not be perceived as 'better' if patient knew of their relative limb volume outcomes compared to CDT or if the therapies were swapped in the same patients?

Minor:

The discussion does make it seem like the effect is more profound and could be tempered slightly.

Line 54: the interstitial fluid in lymphedema is not necessarily any more protein-rich than the contralateral tissue (PMID: 8287667)

Reviewer 3 Report

The authors presented well prepared and performed investigation on the important subject of lymphedema treatment. My concerns are listed below:

  1. Introduction – after reading this chapter readers can have the incomplete view of the current therapeutic options in lymphedema. The authors gave one-sided presentation, of the inadequate value of compression. It would be better present the literature on compression in which quality of life could be improved, thus present pros and cons of compression.
  2. Study participants – Besides clinical stage of the disease the differences between the limbs should be included, e.g. how many of the were in stage one and two.
  3. Procedure – This Antiedema Therapy is a kind of new technique, so it must be well presented, as it can be repeated by other researchers in the future. Instead of full presentation a detailed literature of this method should be given. I cannot find the information on the duration of the treatment
  4. Methods – The new therapeutic technique should be checked in the field of the impact on the size of limb edema.
  5. Follow-up – There is only a short (?) observation performed, which could be seen as the serious problem in assessing quality of life.
  6. Discussion – This chapter should also mention other therapeutic options in lymphedema that may influence quality of life, especially in longer time of observation.
  7. Discussion – The sentence beginning with “The proposed experimental intervention, AAPT, has demonstrated …” is neither based on the results nor the literature given.
  8. Conclusions – Such strict conclusions cannot be taken from this research. The additional benefit suggest that investigators added something new to the main stream methods which is not true. These conclusions, taking into account the absence of the follow up, can mislead readers. In the longer observation in the absence of compression and lymphedema worsening, quality of life may also worsen.

Round 2

Reviewer 1 Report

I deeply appreciate the efforts made by the authors to respond to my comments and to integrate the changes into the document.
In general, the manuscript has improved. However, it still calls my attention that they state that there is no clear treatment or clinical recommendations for the treatment of lymphedema, when in the previous review I have provided that information. If there is evidence and information on that topic, it should be reflected in the document.
On the other hand, it is true that there is controversy about prevention methods for lymphedema. However, I believe that the objective of the study is not its prevention.
In the attached document, you can find my new comments to some of their responses in bold and underlined.

Author Response

Please, see the attached document

Reviewer 3 Report

Fairly improved manuscript, thank you for you job.

I have only few concerns:

1. Were there any differences in edema volume before the treatment between the groups?

2. Data about the limb volumes and edema volumes before and after the treatment should be shown to present if the estimated limb volume reduction was achieved.

Author Response

Please, see the attached document 
